# Prediction of Oleic Acid Content of Rapeseed Using Hyperspectral Technique

**Fan Liu, Fang Wang, Guiping Liao \*, Xin Lu and Jiayi Yang**

Southern Regional Collaborative Innovation Center for Grain and Oil Crops, Hunan Agricultural University, Changsha 410128, China; liuqing@stu.hunau.edu.cn (F.L.); f.Wang@hunau.edu.cn (F.W.); luxin@stu.hunau.edu.cn (X.L.); yangjiayi@stu.hunau.edu.cn (J.Y.)
\* Correspondence: lgpxf@hunau.net; Tel.: +86-151-1636-9008

**Abstract:** In order to detect the oleic acid content of rapeseed quickly and accurately, we propose, in this paper, an artificial BP neural networks based model for predicting oleic acid content by using rapeseed's hyperspectral information. Four types of spectral features are selected for our investigation, namely multifractal index, sensitive band, trilateral parameters, and spectral index. Both univariate variable and multiple variables are considered as our model input. The result shows that the combined feature has higher precision and better stability than when using a single parameter. An interesting finding shows that the combined feature involving multifractal parameters can significantly improve the model performance. Taking the combined feature {MF-$h$(0), SB-DR$_{574}$, SPI-NDSI($R_{575}$, $R_{576}$)} as the model input, the constructed BP (back propagation) neural networks model has the highest precision, with the coefficient of determination ($R^2$) 0.8753, root mean square error (RMSE) 1.0301, and relative error (RE) 1.047%. This result provides some experience for the rapid detection of rapeseed's oleic acid content.

**Keywords:** hyperspectral; multifractal; hurst exponent; oleic acid content; BP neural networks





## 1. Introduction

With the increase of rapeseed oil production in the world, people pay more and more attention on how to improve rape quality as well as to cultivate high oleic rapeseed. The higher the oleic acid content is, the higher the nutritional value and the longer the shelf life will be expected. The oleic acid is an indispensable nutrient element in animal food. It plays a pivotal role in the metabolism of humans and animals. In addition, high oleic sauerkraut oil can effectively prevent human cardiovascular disease. Due to its high economic and nutritional value, evaluation and predictions of the high oleic acid rapeseed have become a hot research area in recent years [1].

The traditional determination of oleic acid content relies on the gas chromatography method, which is time consuming and labor intensive. The prominent disadvantage is the destruction of seeds, which may disqualify the seed from being used used for sowing and reproduction. By this token, this method is not suitable for analyzing and the screening of large quantities of breeding materials of rare and precious quality [2]. Therefore, discovering methods that can quickly and accurately provide diagnosis of the fatty acid content in rapeseed is a critical job for the improvement of rapeseed fatty acid.

The rapid development of hyperspectral technology has created conditions for solving this problem. Scientist apply the hyperspectral technology to measure the seed spectrum information for crop growth diagnosis. The advantages are fast and non-destructive and therefore, in recent years, it has become the important pattern reform for determining the content of oleic acid. This has also attracted many scholars to study hyperspectral technology in crop diagnosis [3–6]. Due to the fact that the hyperspectral imaging technology combines image technology and spectroscopy technology, which can obtain image information and biochemical information of the research object at the same time, it gradually

replaced the traditional infrared spectroscopy technology and has become the main tool in the field of crop growth diagnosis.

Hyperspectral remote sensing data provides rich information of the object; however, it has strong mutuality and redundancy between adjacent bands, which may reduce the efficiency and even accuracy. Hence, it is necessary to extract and analyze the original hyperspectral curve. Among the multitudinous methods of hyperspectral feature analysis, the methods based on mathematical transformation (including statistics, physics, computing, etc.) will provide more robust results [7]. As a powerful and novel mathematical tool, fractal theory is always used to explain the nonlinear and complex structure in nature. In this work, we introduce multifractality into spectral feature analysis due to the fractal nature of the hyperspectral data [8–10]. Since the spectral reflectance curve is not only self-similar but also nonstationary, the application of the multifractal detrended fluctuation analysis (MF-DFA) method into this instance is good choice because this method can deal well with non-stationary measure in various fields [11–14].

On the other hand, the inversion of oleic acid by using Hyperspectral Information depends on high performance training models. BP neural network model is a multi-layer feed forward neural network that is trained based on the error back propagation algorithm. Owing to the powerful ability of nonlinear mapping and prediction, it is widely used in hyperspectral inversion [15]. Among them, Wang et al. [16] compared the BP neural network model and traditional regression model for estimating the accuracy of wheat biomass based on the hyperspectral vegetation index and found that the BP neural network was the champion among those methods. By using BP-based artificial neural network, Yao et al. [17] considered the red-edge parameter as the input variables to estimate the chlorophyll content of the leaves of French platanus and Populus tomentosa. Chen et al. [18] used several characteristic bands, green peaks, and red-edge positions to construct a BP neural network model to invert the pigment content of rice.

In this paper, we propose a spectral inversion model for the forecast of rapeseed's oleic acid content. The hyperspectral information of rapeseed is utilized in our considerations. By using three kinds of traditional hyperspectral characteristics together with its multifractal feature calculated by MF-DFA as model input, the spectral inversion model is constructed based on BP neural networks.

The rest of this paper is organized as follows. In Section 2, we firstly provide a brief account of the experimental materials (Section 2.1), then we review the well-known MF-DFA method and BP neural networks (Section 2.2). In Section 3, we first present four kinds of spectral characteristics (Section 3.1), then we select some of them as model input by using correlation analysis (Section 3.2). Using those selected spectral characteristics as model input, the prediction of rapeseed's oleic acid content is constructed based on BP neural networks (Section 3.3). Finally, we give a brief summary in Section 4.

## 2. Materials and Methods

### 2.1. Materials

The Selected Materials

Two rapeseed varieties (Xiangyou 708 and Xiangyou 710) are used as the research objects. Our field experiment is located in the Yunyuan Base of Hunan Agricultural University ($113°4'$ E, $28°10'$ N) in 2018 and 2019. The field soil is black loam soil with rice-rapeseed rotation.

### 2.2. Data Collection

The SOC710 portable hyperspectral imager (wavelength range 400–1000 nm, resolution 4.6875 nm), produced by Surface Optics Corporation (11555 Rancho Bernardo Road San Diego, CA 92127, USA) of the United States, is used for spectrum measurement. The dark box of the optical experiment (task cabin) is placed in dark room conditions. The bottom area of the box is 50 * 60 cm and the height is 100 cm with a movable base. The interior of the dark box has a diffuse reflection coating. Four sets of surround-type built

with 70 W halogen light source with an incident angle 45° was used. It has a cooling device (in accordance with the requirements of the SOC710 hyperspectral imaging spectrometer). The instrument is placed vertically and directly above the target with the exposure distance of 300 mm, preheated for 15 min, and then used to measure spectral information. We put the rape seeds in 16 circular dishes. In every dish, 5 non-overlapping rectangular areas (see Figure 1) are randomly selected as the region of interest (ROI) and then 80 ROIs can be obtained. In each ROI, we randomly measure the spectral reflectivity five times and average them as the final spectral reflectivity value of the ROI. In this manner, 80 spectral reflectivity can be obtained that is labelled as 1~80. Table 1 shows the statistical characteristics of rapeseed oleic acid content.

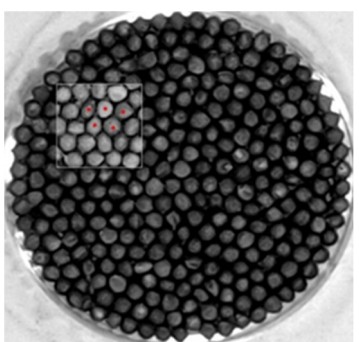

**Figure 1.** ROI of rapeseed sample.

**Table 1.** Statistical characteristic of oleic acid content rapeseed samples.

| Number | Min/% | Max/% | Mean/% | Standard Derivation/% | Coefficient of Variation |
|:------:|:-----:|:-----:|:------:|:---------------------:|:------------------------:|
| 80 | 65.827 | 80.085 | 75.176 | 3.753 | 0.0499 |

On the other hand, all of the rapeseeds in every ROI are grinded to measure the fatty acid content of rapeseed and to further to obtain rapeseed oleic acid data by using Agilent 7890 Inductively Coupled Plasma Mass Spectrometer in the Oil Research Institute of Hunan Agricultural University, where the indoor temperature is 16 °C and the relative humidity is 40%. The experimental environment is as follows:

- Chromatography column: DB-5 Chromatography column (30 m × 0.25 mm × 0.25 μm, 5% Phenyl-95% Polydimethylsiloxane);
- Temperature programmed condition: The initial temperature is 50°. The temperature is then raised to 200 °C at 10 °C/min and held there for 10 min. The temperature is then raised to 300 °C at 10 °C/min and held there for 10 min again;
- Vaporization chamber temperature: 300 °C;
- Detector temperature: 300 °C;
- Carrier gas: high-purity nitrogen, gas flow ratio 100:1;
- Column flow rate: 1 mL/min;
- Cracking furnace temperature: 350 °C;
- Interface temperature: 300 °C;
- Fatty acid methyl esterification: 400 mL potassium hydroxide methanol solution and 800 mL ether petroleum ether solution.

From the above process, we obtained 80 samples of spectral reflectivity and oleic acid data, out of which 64 were used as the training samples and the leftovers were used as the test samples.

### 2.3. Method

2.3.1. Multifractal Detrended Fluctuation Analysis (MF-DFA)

For a given hyperspectral reflectance series $\{x_i\}_{i=1}^{N}$, we calculated the profile as $\{X_t\} = \sum_{i=1}^{t}(x_i - \overline{x})$, where $\overline{x}$ is the average of $\{x_i\}_{i=1}^{N}$ over the band length $N$. Divide

the profiles $\{x_t\}_{t=1}^N$ into $Ns \equiv [N/s]$ non-overlapping segments with equal length $s$. A short part of data at the end of profiles could be left, since that the $N$ is not always an integral multiple of the given scale $s$. In order to prevent the loss of original information, the dividing procedure is repeated starting from the opposite end of the profile. Thus, a total of $2Ns$ segments is obtained. Accordingly, the $v$-th segment is denoted as $[lv + 1, lv + s]$, where $lv = (v - 1)s$ for $v = 1, 2, \ldots, Ns$ and $lv = N - (v - Ns)s$ for $v = Ns + 1, Ns + 2, \ldots, 2Ns$.

Next, in each segment $v$, determine the local trend by using polynomial fitting $\widetilde{y}_v(k)$. Denote $y_s(k)$ by the detrended series in $v$-th segment, as shown in the following.

$$y_s(k) = y(k) - \widetilde{y}_v(k), \; k = 1, 2, \cdots, N_s \tag{1}$$

In this work, we use a first order polynomial to fit the trend. Following this, we calculated the variance for every detrended series in segment $v$.

$$F^2(s, v) = \begin{cases} \frac{1}{s} \sum_{i=1}^s y_s^2[(v-1)s + i], & v = 1, 2, \ldots, N_s, \\ \frac{1}{s} \sum_{i=1}^s y_s^2[N - (v - N_s)s + i], & v = N_s + 1, N_s + 2, \ldots, 2N_s. \end{cases} \tag{2}$$

In addition, the averaged $q$-order fluctuation function ($q \neq 0$) over the all the segment can be calculated according to the following:

$$F_q(s) = \left[ \frac{1}{2N_s} \sum_{v=1}^{2N_s} \left[ F^2(s, v) \right]^{q/2} \right]^{1/q} \tag{3}$$

and when $q = 0$, according to $L'$ *Hospital*, the $F_q(s)$ is determined by the following.

$$F_0(s) = \exp\left[ \frac{1}{4N_s} \sum_{v=1}^{2N_s} \ln F^2(s, v) \right] \tag{4}$$

Finally, vary the scale $s$ and repeat Equations (1)–(4) to calculate the corresponding $q$-th fluctuation function $F_q(s)$. If the spectrum possesses fractal nature, there exists a power-law scaling behavior between the $F_q(s)$ and $s$ as described in the following.

$$F_q(s) \propto s^{h(q)} \tag{5}$$

The index $h(q)$ can be obtained by the linear fitting of $F_q(s)$ and $s$ in a double-log plot. The $h(q)$ is called generalized MF-st exponent and describes the long-term correlation for the original spectrum. Generally, $h(q) > 0.5$ expresses persistence of the spectrum reflectance series $\{x_i\}_{i=1}^N$ and the $h(q) < 0.5$ is anti-persistent. The multifractal nature is present in case of dependence of $h(q)$ on $q$. In order to measure the degree of multifractality, the $\Delta h$ defined by Equation (6):

$$\Delta h = h_{max}(q) - h_{min}(q) \tag{6}$$

where $h_{max}(q)$ and $h_{min}(q)$ are the maximum and minimum of the $h(q)$ for the considering $q$s, respectively. In this work, we took $q \in [-3, 3]$. The larger $\Delta h(q)$ is, the higher the strength of multifractality is expected to be.

According to the typical multifractal analysis (MFA), the quality index $\tau(q)$, which can also express the multifractal nature, is related with $h(q)$ as the following:

$$\tau(q) = qh(q) - D_f \tag{7}$$

where $D_f$ is the topological dimension of the object. For the spectral reflectance series, the $D_f = 1$. When the $\tau(q)$ is the nonlinear function of $q$, the multifractality of object can be observed.

Via the *Legendre* transformation, the *Lipschitz–Hölder* index $\alpha(q)$ and multifractal spectrum $f(\alpha)$ are determined by the following.

$$\begin{cases} \alpha(q) = \tau'(q) = h(q) + qh'(q) \\ \quad f(\alpha) = q\alpha(q) - \tau(q) \end{cases} \tag{8}$$

$$\Delta\alpha = \alpha_{max}(q) - \alpha_{min}(q) \tag{9}$$

In practice, the $\Delta\alpha$ is the span of the multifractal spectrum. The larger $\Delta\alpha$ is, the more uneven the reflectance distribution is and the greater the fluctuation is observed. In this work, above 12 multifractal parameters are employed as augments for our consideration, namely $h(\pm 3)$, $h(\pm 2)$, $h(\pm 1)$, $h(0)$, $\Delta h$, $\alpha_{max}$, $a_{min}$, and $\Delta\alpha$.

2.3.2. BP Neural Networks

BP (back propagation) neural network is a concept put forward by Rumelhart and McClelland in 1986 [19]. It is a multilayer feedforward neural network trained according to the error back propagation algorithm. The structure of BP neural network contains input layer, hidden layer, and output layer, out of which, there are one or more layers in the hidden layer. Each neuron in two adjacent layers is connected to all neurons, while there is no connection between neurons in the same layer. In this manner, the BP neural network can deal with more complex computational tasks. Since the BP neural network has the back-propagation mechanism, the mean square of the difference between the actual output and the expected output can be regarded as an error signal to propagate back along the network in supervised learning. During the propagation process, the weight of each layer will be adjusted. This process ends when the error is lower than the target value [20]. In this work, we utilize the BP neural network to construct oleic acid content prediction model.

For the 80 groups of rapeseed samples, the hyperspectral features are regarded as the input layer, while the oleic acid content of rapeseed is used as the output layer in BP neural network. The number of hidden layer nodes $p$ is based on the range given by the empirical Equation (10). By using a trial-and-error method for multiple training (100 training times for each sample), the optimal number of nodes is 9.

$$p = \sqrt{k+m} + a \tag{10}$$

$k$ is the number of input layer units; $p$ is the number of hidden layer nodes; $m$ is the number of output layer units; and $a$ is the constant of 1–10. Set the number of iterations as 1000 and the learning accuracy as 0.01. The 64 samples are then randomly selected as the training set and the leftover 16 samples are regarded as the test set. By using the Trainlm training method [21] with cross-validation, the BP network predication model can be constructed and optimized to select the best model parameters.

*2.4. Evaluate Indicator*

In order to evaluate the model performance, three indicators, namely the coefficient of determination($R^2$), root mean square error (RMSE), and relative error (RE), are employed in this work and shown as follows:

$$R^2 = \frac{\sum_{i=1}^{n}\left(\hat{Y}_i - \overline{Y_p}\right)^2}{\sum_{i=1}^{n}\left(Y_i - \overline{Y}\right)^2} \tag{11}$$

$$\text{RMSE} = \sqrt{\frac{1}{n}\sum_{i=1}^{n}\left(\hat{Y}_i - Y_i\right)^2} \tag{12}$$

$$\text{RE} = \frac{1}{n}\sum_{i=1}^{n}\left|\frac{\hat{Y}_i - Y_i}{Y_i}\right| \times 100\% \tag{13}$$

where $Y_i$ is the observed value, $\hat{Y}_i$ is the predicted value, $\overline{Y}$ is the average observed value, $\overline{Y_p}$ is the average predicted value, and $n$ is the total number of samples. The three evaluators describe the model's interpretation ability, model error, and model relative error, respectively.

## 3. Result and Discussion

### 3.1. Feature Selection

The red-edge parameter is one of the most significant characteristics of the green plant spectrum. It refers to the spectral position (wavelength) corresponding to the maximum value of the first derivative spectrum in the red-light range (680~760 nm). The red-edge amplitude refers to the maximum value of the first derivative spectrum in the red-light range. The red-edge area is the integral of the first derivative in the red-light range. Similarly, the blue-edge (490~530 nm) parameter and the yellow-edge parameter (560~640 nm) are also regarded as important features of the green plant. They are collectively called trilateral parameter (TriP) [22].

Spectral index is a linear or nonlinear combination of spectral reflectance at some specific bands, which has a certain meaning for the object. Generally, the Ratio spectral index (RSI), Normalized difference spectral index (NDSI), and Difference spectral index (DSI) are the three most significant spectral indexes, which are selected to our consideration and shown in Equations (14)–(16) ($R_{\lambda_1}$ and $R_{\lambda_2}$ denote the reflectance of the wavelength $\lambda_1$ and $\lambda_2$, respectively). The hyperspectral parameters mentioned in this paper are summarized in Table 2.

$$\text{RSI}\,(\lambda_1,\,\lambda_2) = \frac{R_{\lambda_1}}{R_{\lambda_2}} \tag{14}$$

$$\text{NDSI}\,(\lambda_1,\,\lambda_2) = \frac{R_{\lambda_1} - R_{\lambda_2}}{R_{\lambda_1} + R_{\lambda_2}} \tag{15}$$

$$\text{RSI}\,(\lambda_1,\,\lambda_2) = R_{\lambda_1} - R_{\lambda_2} \tag{16}$$

**Table 2.** Hyperspectral characteristic parameters.

| Parameters | Symbol | Description | Ref |
|---|---|---|---|
| Multifractal feature | MF-$h(0)$ | Generalized Hurst exponent | [8–10] |
| | MF-$h(\pm 1)$ | Generalized Hurst exponent | [8–10] |
| | MF-$h(\pm 2)$ | Generalized Hurst exponent | [8–10] |
| | MF-$h(\pm 3)$ | Generalized Hurst exponent | [8–10] |
| | MF-$\Delta h$ | Span of h(q) | [8–10] |
| | MF-$\alpha_{max}$ | Maximum of *Hölder* index | [8–10] |
| | MF-$\alpha_{min}$ | Minimum of *Hölder* index | [8–10] |
| | MF-$\Delta \alpha$ | Span of $\alpha(q)$ | [8–10] |
| Sensitive band | SB-R | Sensitive band of reflectance | [23] |
| | SB-DR | Sensitive band of the first derivative reflectance | [23] |
| Trilateral Parameter | TriP-rep | Red edge position | [24] |
| | TriP-Dr | Red edge amplitude | [24] |
| | TriP-SDr | Red edge area | [24] |
| | TriP-yep | Yellow edge position | [25] |
| | TriP-Dy | Yellow edge amplitude | [25] |
| | TriP-SDy | Yellow edge area | [22] |
| | TriP-bep | Blue edge position | [25] |
| | TriP-Db | Blue edge amplitude | [25] |
| | TriP-SDb | Blue edge area | [22] |
| Spectral index | SPI-RSI | Ratio spectral index | [26] |
| | SPI-DSI | Difference spectral index | [27] |
| | SPI-NDSI | Normalized difference spectral index | [28] |

The multifractal feature captures the global singularity and correlation of the hyperspectral reflectance, which may reflect the essential characteristics of the spectral reflectance of rapeseed samples with different oleic acid content. The spectral index expresses the combined characteristics of the reflectance at different bands. The trilateral parameter focuses on the hyperspectral characteristics change at special locations. The sensitive band locates the band where the hyperspectral reflectance has the most significant correlation. In the following, we use these four types of parameters as features to predict the oleic acid content of rapeseed.

### 3.2. Correlation Analysis of Spectral Parameters and Oleic Acid Content

In order to choose the best parameters as the argument model for the four types of hyperspectral parameters mentioned above, we conducted a correlation analysis for the oleic acid content of rapeseed with all the parameters and reported the results in Figure 2. As shown in those subplots, for the multifractal features, $h(0)$ possessed the best correlation coefficient 0.7898 and is superior to other Hurst exponents. For the trilateral parameters, the best correlation coefficient 0.7751 comes from the area of the yellow edge. In subplot Figure 2c, the original reflectance (blue line) and first-order derivative reflectance (red line) shows different performance. The maximum of the correlation coefficient is 0.782 and the corresponding wavelength is at 574 nm. The correlation coefficients between the rapeseed oleic acid content and the three spectral indexes of RSI, NDSI, and DSI are shown in subplot Figure 2d–f, respectively. By comparison, NDSI brings the best result with correlation coefficient being 0.7950 and the corresponding optimal spectral index is NDSI ($R_{575}$ and $R_{576}$).

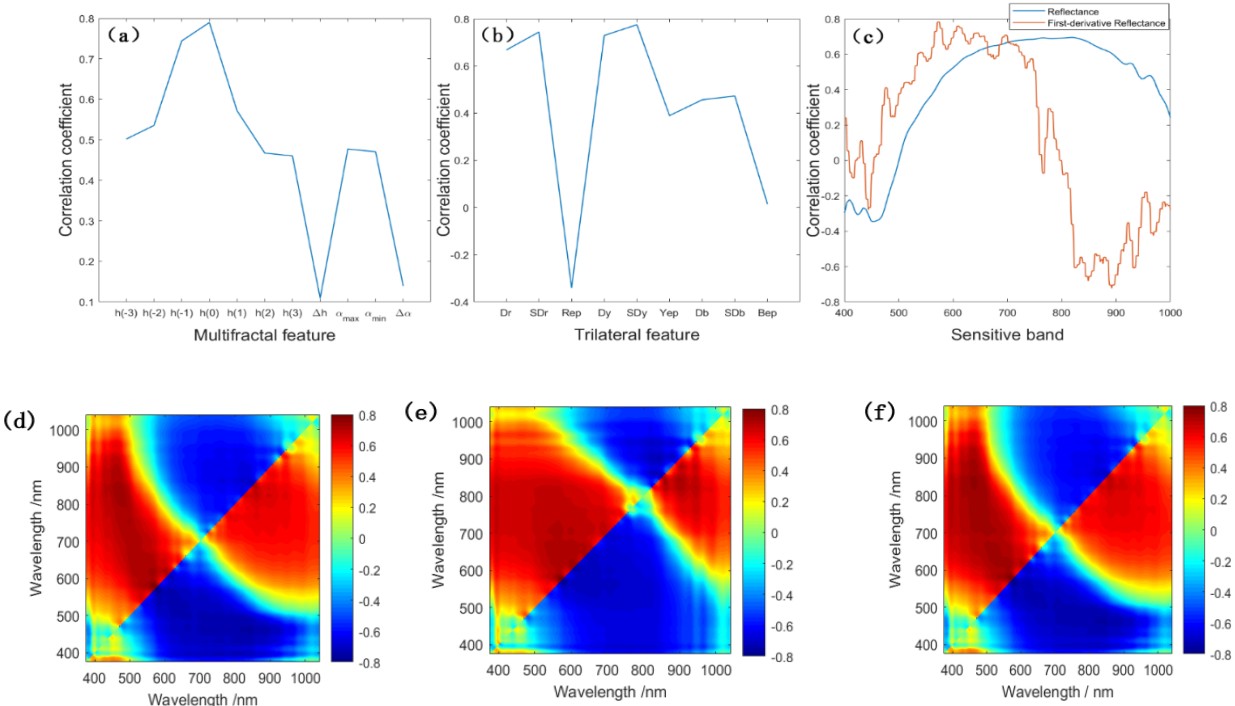

**Figure 2.** Correlation analysis between four types of parameters and the oleic acid content. (**a**–**c**) are the correlation coefficient of the oleic acid content with respect to the multifractal feature, trilateral feature, and sensitive band, respectively. (**d**–**f**) represent the correlation coefficient between the oleic acid content and three spectral indexes, namely the ratio spectral index (RSI), normalized spectral index (NDSI), and difference spectral index (DSI).

According to the above correlation analysis, we select the first two parameters with the best correlation in each type of hyperspectral feature, as listed in Table 3. The eight parameters with the higher correlation coefficients are greater than 0.7, which passes the correlation test under the 0.01 significance level. Table 4 shows the statistics of the eight

selected spectral feature. In the following, we use the above eight characteristic parameters to establish an estimation model for oleic acid content with BP neural network.

**Table 3.** The first two parameters with the best correlation in each hyperspectral feature.

| Parameter | Correlation Coefficient |
|---|---|
| MF-$h(0)$ | 0.7898 ** |
| MF-$h(1)$ | 0.7442 ** |
| TriP-SDy | 0.7751 ** |
| TriP-SDr | 0.7442 ** |
| SPI-DSI ($R_{572}$, $R_{574}$) | 0.7862 ** |
| SPI-NDSI ($R_{575}$, $R_{576}$) | 0.7950 ** |
| SB-$R_{818}$ | 0.7022 ** |
| SB-DR$_{574}$ | 0.782 ** |

** 0.01 significance level.

**Table 4.** The statistics of the eight selected spectral feature.

| | Min | Max | Mean | STDEV |
|---|---|---|---|---|
| MF-$h(0)$ | 1.1976 | 1.8628 | 1.5511 | 0.1269 |
| MF-$h(1)$ | 1.2306 | 2.0727 | 1.5900 | 0.1993 |
| TriP-SDy | 0.0176 | 0.5693 | 0.2328 | 0.1490 |
| TriP-SDr | 0.3019 | 1.0011 | 0.6179 | 0.1812 |
| SPI-DSI ($R_{572}$, $R_{574}$) | −0.0068 | 0.0118 | 0.0037 | 0.0037 |
| SPI-NDSI ($R_{575}$, $R_{576}$) | −0.0012 | 0.0047 | 0.0021 | 0.0017 |
| SB-$R_{818}$ | 0.2867 | 0.8076 | 0.4937 | 0.1308 |
| SB-DR$_{574}$ | −0.0002 | 0.0015 | 0.0005 | 0.0004 |

### 3.3. Estimation Model of Oleic Acid Content

The above four types of spectral feature MF-◇, SB-◇, TriP-◇, and SPI-◇ are combined as the input layer of BP neural network model, meanwhile the oleic acid content of rapeseed is combined as the output layer. The symbol '◇' in MF-◇, SB-◇, TriP-◇, and SPI-◇ denotes $h(0)$ and $h(1)$, $R_{818}$ and DR$_{574}$, SDy and SDr, and DSI ($R_{572}$, $R_{574}$) and NDSI ($R_{575}$, $R_{576}$), respectively.

According to the number of characteristic parameters in the combination, univariate, bivariate, ternary, and quaternary models are constructed (At most, only one of each type of feature is selected for combination feature). In this manner, 8 univariate combinations, 24 bivariate combinations, 32 ternary combination, and 16 quaternary combinations can be obtained. We then used the $R^2$, RMSE, and RE to evaluate the performance of BP-based neural network model. Since the result of the BP-based algorithm depends on the random initial weight, the modelling process is repeated 100 times and averaged for comparison. Figure 3 shows the average of $R^2$ over all possible combinations for the univariate, bivariate, ternary, and quaternary models. It clearly shows that the $R^2$ increases (meanwhile the error-bar is decreasing) with increasing the number of parameters in the combination feature. Table 5 lists the best model performance of the four corresponding combinations for the training set and testing set, respectively. An interesting finding uncovers that the performance obtained from the multivariate combination is significantly better than that of the univariate.

According to the Table 5, the model performance obtained from the ternary combination and quaternary combinations is significantly superior to that of the univariate and slightly better than that of bivariate combination. Figures 4 and 5 show the visual model results for the training set (left panel) and testing set (right panel) by using the ternary combination {MF-$h(0)$, SB-DR$_{574}$, SPI-NDSI($R_{575}$, $R_{576}$)} and quaternary combination {MF-$h(0)$, SB-DR$_{574}$, TriP-SDr, SPI-NDSI ($R_{575}$, $R_{576}$)}, respectively.

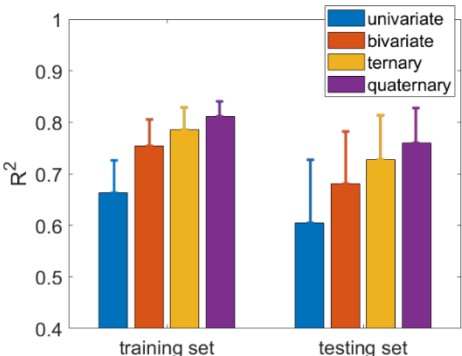

**Figure 3.** Average of $R^2$ for the combinations containing different variables.

**Table 5.** The best model of different parameter combinations.

| | **TRAIN ($N = 64$)** | | | **TEST ($N = 16$)** | | |
|---|---|---|---|---|---|---|
| | $R^2$ | RMSE | RE (%) | $R^2$ | RMSE | RE (%) |
| {SPI-NDSI($R_{575}$, $R_{576}$)} | 0.7777 | 1.3647 | 1.377 | 0.6799 | 1.55 | 1.728 |
| {MF-$h$(0), SB-DR$_{574}$} | 0.8635 | 1.0603 | 1.128 | 0.7971 | 1.3388 | 1.678 |
| {MF-$h$(0), SB-DR$_{574}$, SPI-NDSI($R_{575}$, $R_{576}$)} | 0.8753 | 1.0301 | 1.047 | 0.8169 | 1.2922 | 1.497 |
| {MF-$h$(0), SB-DR$_{574}$, TriP-SDr, SPI-NDSI($R_{575}$, $R_{576}$)} | 0.8652 | 1.0486 | 1.085 | 0.8012 | 1.2706 | 1.526 |

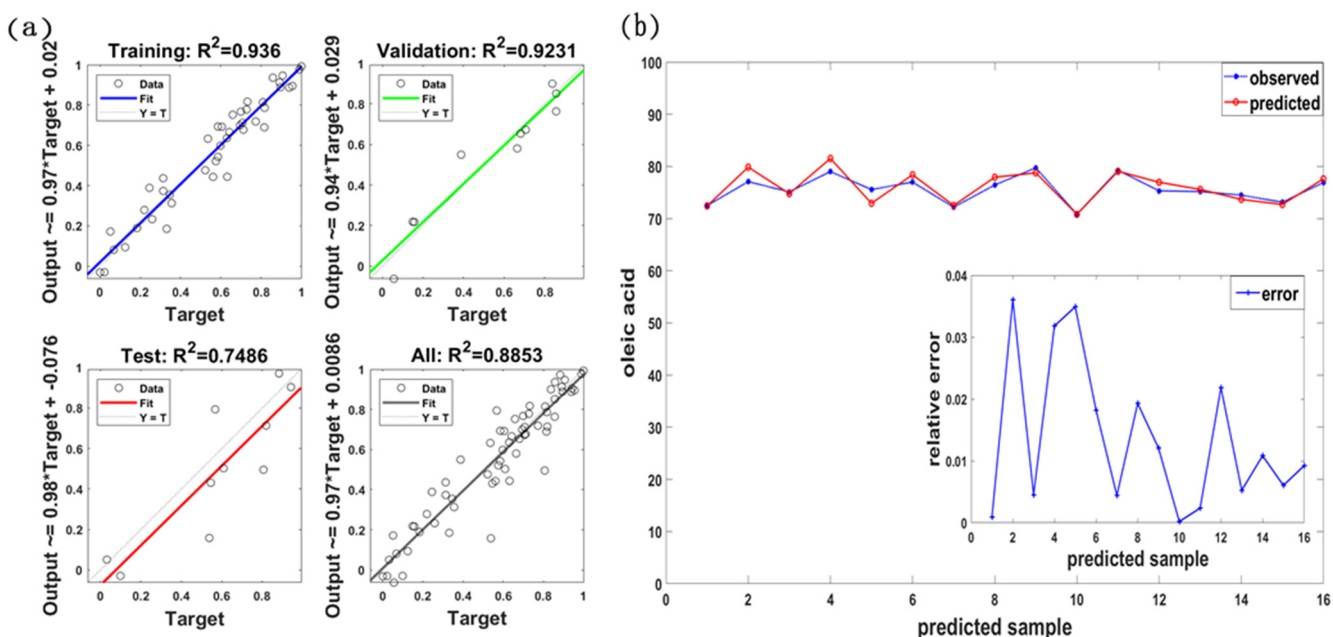

**Figure 4.** Model results obtained from the ternary combination {MF-$h$(0), SB-DR$_{574}$, SPI-NDSI($R_{575}$, $R_{576}$)}. (**a**) is for the 64 training samples. (**b**) is for the 16 testing samples.

In addition, as mentioned in Section 3.1, the multifractal feature depicts the global characteristic of the hyperspectral reflectance, which may bring better model performance for predicting rapeseed's oleic acid content. In order to investigate this, we compared the model results obtained between the feature combinations including and excluding the Hurst exponent. Here, we considered the combinations of univariate, bivariate, and ternary cases. For example, for the bivariate combination, there are 12 combinations including MF parameters ($h$(0) and $h$(1)) and other 12 combinations exclude them. The averaged

results over the all-possible combinations are listed in Table 6. It can be observed from the results that the Hurst exponent is not as good as the traditional spectral parameters when the univariate is considered as the argument. However, the Hurst exponent exhibits its superiority in the case of the multivariate model because it brings a significantly better model performance. This finding suggests that the multifractal feature should be an important supplement to the traditional spectral characteristics when we construct the rapeseed's oleic acid content evaluation model.

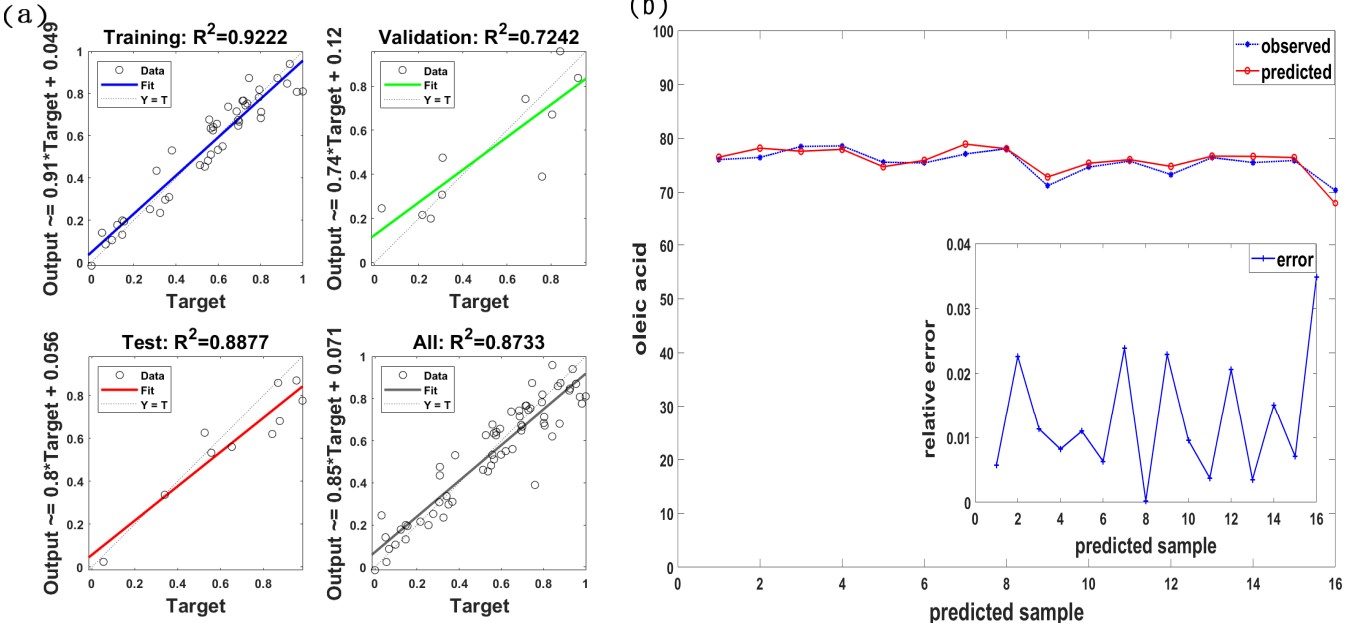

**Figure 5.** Model results obtained from the quaternary combination {MF-$h$(0), SB-DR$_{574}$, TriP-SDr, SPI-NDSI($R_{575}$, $R_{576}$)}. (**a**) is for the 64 training samples. (**b**) is for the 16 testing samples.

**Table 6.** The influence of MF parameters on model accuracy.

| | MF-Features | Training Set ($N$ = 64) | | | Testing Set ($N$ = 16) | | |
|---|---|---|---|---|---|---|---|
| | | $R^2$ | RMSE | RE (%) | $R^2$ | RMSE | RE (%) |
| Univariate | with | 0.6232 | 1.748 | 1.913 | 0.6043 | 1.8096 | 2.154 |
| | without | 0.6789 | 1.6399 | 1.752 | 0.6201 | 1.8153 | 2.086 |
| Bivariate | with | 0.7835 | 1.3451 | 1.432 | 0.7152 | 1.5610 | 1.816 |
| | without | 0.7122 | 1.5326 | 1.634 | 0.6421 | 1.7567 | 2.274 |
| Ternary | with | 0.8086 | 1.2498 | 1.353 | 0.7573 | 1.4679 | 1.685 |
| | without | 0.7394 | 1.4754 | 1.601 | 0.6685 | 1.6988 | 1.983 |

As the last important task, the model test will show the model stability. In order to perform this, we test the model by changing the number of training samples. According to the parameter combinations listed in Table 5, 48–72 samples are randomly selected as the training set, the three evaluators of $R^2$, RMSE, and RE are calculated and shown in Figure 6. It is clearly shown that there is non-significant change of the three indicators with the increasing training numbers, which implies that the selected feature combination brings stable model result.

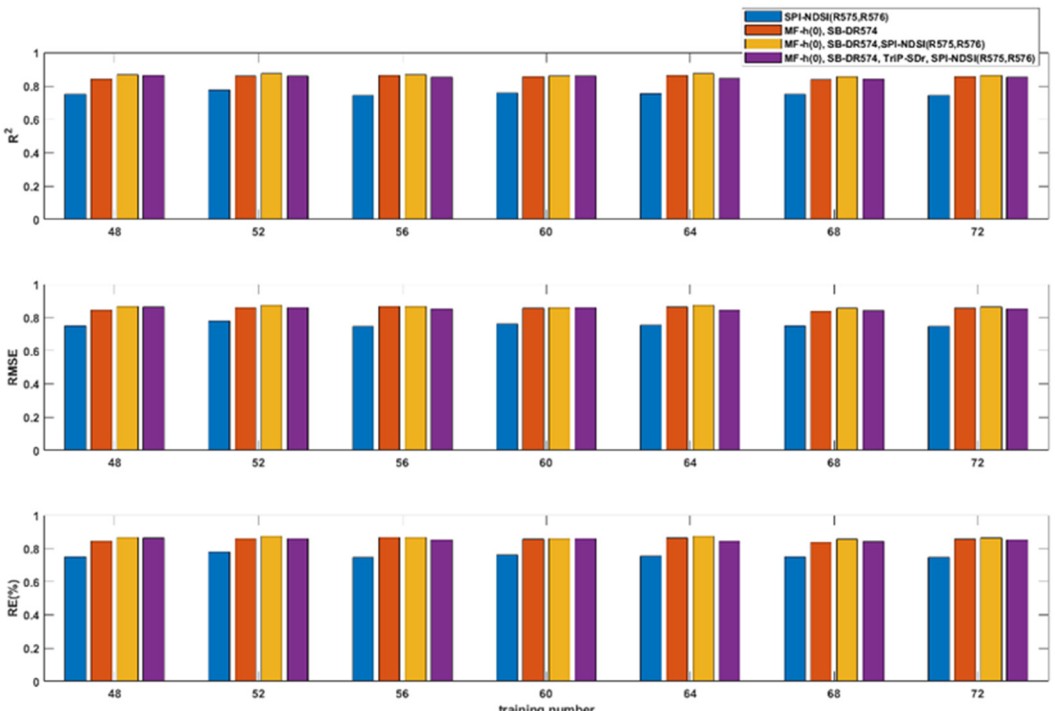

**Figure 6.** Three evaluation indicators of *RMSE*, *R*, and *RE* with different sample sizes of training set.

## 4. Conclusions

Hyperspectral technology possesses the advantages of being fast, non-destructive, and highly efficient and, therefore, it can play an important role in crop nutrition diagnosis. In this paper, we attempt to use the hyperspectral characteristics of seeds to construct the inversion model of rapeseed's oleic acid content. The proposed inversion model provides a helpful experience for estimating the oleic acid non-destructively. In practice, based on rapeseed hyperspectral data, four types of spectral features are considered, namely multifractal parameters, trilateral parameters, spectral indices, and sensitive bands. In order to select optimal characteristic parameters, we first choose two features in each type of spectral features according to correlation analysis. Then, by using the selected features as the model input of univariate (one feature) and multivariate combination (at least two features), the BP neural network model is established for an oleic acid prediction.

The results show that multivariate parameters can greatly improve the model's accuracy and stability. An interesting finding shows that the combined features including multifractal parameters will bring about better model performance. The best model performance comes from the combined parameters $\{MF\text{-}h(0), SB\text{-}DR_{574}, SPI\text{-}NDSI(R_{575}, R_{576})\}$. Model test shows that our model has good robustness.

**Author Contributions:** Conceptualization, F.L.; methodology, F.L. and G.L.; software, F.L.; validation, F.L.; formal Analysis, F.L. and F.W.; investigation, F.L. and X.L.; resources, G.L.; data curation, F.L.; writing—Original Draft Preparation, F.L.; writing—review and editing, F.W. and J.Y.; visualization, F.L. and X.L.; supervision, G.L.; project administration, G.L.; funding acquisition, G.L. All authors have read and agreed to the published version of the manuscript.

**Funding:** This research was funded by the Chinese National Natural Science Foundation (Grant No. 61973111) and the Natural Science Foundation of Hunan Province (CN) (Grant No. 2020JJ4377).

**Data Availability Statement:** The raw/processed data required to reproduce these findings cannot be shared at this time as the data also forms part of an ongoing study.

**Conflicts of Interest:** The authors declare no conflict of interest.

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
