# Peer review of "Prediction of Oleic Acid Content of Rapeseed Using Hyperspectral Technique"

_applsci, doi:10.3390/app11125726_

Round 1
Reviewer 1 Report
Thank you very much for submitting your article. Although I consider this article is appropiate to its publication, some key points must be improved. Related to the introduction, I think this is the main point to keep the reader's attetion and the objective of the study must be clear. In this case, the introduction does not reflect the importance or necessity of this work. In the case of the methodology, the sample set used is not enough to develop a complex model such as ANN. I think it would be a good point to consider to try easier chemometric treatments.
Please, try to use impersonal verbs and avoid the use of 'we' or 'us'.

Author Response
Reply to Reviewer’s Comments
We are very grateful to the anonymous reviewers for their valuable comments and suggestions. Based on their comments and suggestions, we have revised our manuscript. In the following we list comments (labeled as C#) from reviewers and give our responses (labeled as A#) right after each comment.
Reply to Reviewer #1
C1. In order to make easier for the reader to keep the importance of the article, it would be interested to provide some information about the amount of the oleic acid in this kind of fruit and about the importance of its consumption. Does the use of the rseeds have any interest for the industry? Furthermore, I am not sure if it makes really sense to talk about the importance of the oleic acid for the human health since the ingested amount is not really high.
A1. Oleic acid is the main fatty acid in rapeseed oil. Its oxidative stability is higher than that of polyunsaturated fatty acids. It is not easy to oxidize and deteriorate at higher temperatures, which helps to extend the shelf life of rapeseed oil. On the one hand, oleic acid is an indispensable nutrient for animals. It plays an important role in the metabolism of human and animals in food. Since that the synthetic oleic acid can't meet demand, a certain amount must be taken from the daily food and rapeseed is an important source of oleic acid. In this regards, the oleic acid content is one of the key factors to measure the quality of rapeseed. On the other hand, oleic acid also plays an important role in industry. It is used to prepare antistatic agents and lubricants in the wool textile industry. It is also used to prepare water repellent paraffin emulsions in the wood industry. Its alcohol solution can be used as rust remover. What’s more, It can be used as agricultural emulsifier, lubricant, printing and dyeing auxiliary, industrial solvent, and metal mineral flotation agent, etc.
We highlight some of those function in our revised version.
C2. I think that the introduction is an important part of the article to focus the reader on the topic. In this case, It would be really interesting to give some notes about the importance of the oleic acid.
A2. This is a very important point. We thank the reviewer to point this. As mention in A1, we’ve added some sentences about the function of oleic acid in Introduction part.
C3. Please, could you specify in the text more about the samples used? Number of each category, for example
A3. We add some descriptions of the rapeseed samples in the end of subsection 2.1 (see lines 126-127)
C4. It should be clearer to resume the experimental environment related to the chromatography in a paragraph.
A4. We add some descriptions of the experimental environment in lines 110-112.
C5. The paragraph related to the ROI is not completely clear. ‘We measure the spectrum on the surface of the repaseed as the ROI’?
A5. We put the rape seeds in 16 circular dishes. In every dish, 5 non-overlapping rectangular areas are randomly selected as the region of interest (ROI), then 80 ROIs can be obtained. For each ROI, measure the spectrum five times and average them as the final spectral value of the ROI. Meanwhile, all of the seeds in the ROI are grinded to measure the oleic acid data. We also reword this in our revised version.
C6. Revise the tittle 2.3
A6. We revised the title 2.3 as “2.3. Evaluate indicator”
C7. I do not really understand the use of the fractal analysis in this work, please could you provide more information about this?
A7. Based on the previous works for rapeseed growth diagnose research, Wang et al[8], Jiang et al [9-10] found that the rapeseed’s hyperspectral shows significant multifractal nature and the fractal features helps to improve prediction accuracy for rapeseed’s chlorophyll. Inspired by their works, we attempt to utilize the fractal features of rapeseed’s hyperspectral together with some typical spectral feature to forecast oleic acid. As shown in Table 6, based on the same Bp model, the combined feature with fractal parameters can greatly improve the model stability and accuracy.
C8. Figure 4. Please increase the size. In the case of the figure 4 and 5, It would be interesting to see the Predicted vs. reference values, as same as for the calibration. Furthermore, I would like to see the SEL for this parameter.
A8. Thanks to the suggestion. We have increased the size of Figures 4 and 5. The selected parameters in Figures 4 and 5 are {MF-h(0), SB-DR574, SPI-NDSI(R575,R576)} and {MF-h(0), SB-DR574,TriP-SDr,SPI-NDSI(R575,R576)}, respectively, which are also shown in the caption.
C9. Applying ANN to a sample set composed just by 80 samples is not appropriate. Have you tried any other models? Could you try a fast PLS to see the results? I think that the easiest, the best…
A9. We thank the reviewer to point this doubt. According to reviewer’s suggestion, we attempt the PLS method to redo our model, the result is listed in below. As shown in Table A, the Bp model with three-combined features brings better model performance than PLS model.
Table A Comparison result of PLS model and BP neural network model
|
|
TRAIN(N=64) |
|
TEST(N=16) |
||||
|
|
R2 |
RMSE |
RE(%) |
|
R2 |
RMSE |
RE(%) |
|
PLS (8 characteristic parameters) |
0.7514 |
1.4271 |
1.52 |
|
0.7490 |
1.4884 |
1.62 |
|
BP{MF-h(0), SB-DR574, SPI-NDSI(R575, R576)} |
0.8753 |
1.0301 |
1.047 |
|
0.8169 |
1.2922 |
1.497 |
C10. To my mind, I think that a work with 80 samples can be considered just a preliminary study and not is enough to talk about nutrition diagnosis. Please, try to modify the beginning of the conclusions.
A10. This is a very constructive comment. We modify some sentences in the beginning of the conclusions.

Reviewer 2 Report
See attached Word file for information.

Author Response
Reply to Reviewer’s Comments
We are very grateful to the anonymous reviewers for their valuable comments and suggestions. Based on their comments and suggestions, we have revised our manuscript. In the following we list comments (labeled as C#) from reviewers and give our responses (labeled as A#) right after each comment.
Reply to Reviewer #2
General comments:
C1. The worry is that the estimates of parameters associated with each of the four types of spectral features may vary with seed maturity stages, with different varieties or with different growing conditions (such as fertiliser levels, soil type, weather) of the same variety.
A1. We thank the reviewer’s doubt. In this work, the rapeseeds we used are collected from the pod, which is at the final growth period of rape. We focus on the relationship between the hyperspectral and its oleic acid. The oleic acid depends only on the final collected rapeseed’s spectral characteristics. The purpose of this work is constructing an experience model to forecast oleic acid by using the spectral information, which is helpful to the nondestructive prediction of oleic acid.
C2. It also needs to be pointed out that how many samples/measurements of spectral features are required to produce the reliable quantification of oleic acid content in the seeds. This is because the values of R2 were different due to sample size differences in the training set and testing set data.
A2. This is very important point. In theory, the more samples, the better performance of the model will be expected. In this work, however, the training samples are set from 48 to 72 (Pls. see Figure 6), we found the model performance keeps stable, which is explained that our model has good robustness.
Minor comments:
C3. PG1 L17 “…the constructed BP neural networks model…”. Give the full name of BP as “BP (back propagation)” here. So, readers will know what it stands for when it appears first time in the manuscript.
A3: Thanks to the reviewer to point it out. Revised as suggested.
C4. PG2 L62 “Of which, …Wang et al….”. It should be “Among them,…”. Wang et al. should be followed by a reference number.
A4: Revised as suggested.
C5. PG4 L144 “In generally, …” it should be corrected to either “Generally” or “In general”.
A5: Revised as suggested.
C6. PG6 L207-208 Give formula that is used to calculate spectral index of RSI, NDSI and DSI in materials and methods.
A6: We thank the reviewer to point this. We have given the formulas of RSI, NDSI and DSI in Eqs. (14)-(16), respectively, of the revision.
C7. PG 8 Table 4. It should be useful to give the parameter estimate of the included spectral feature variables. Other future researchers can then compare their parameter estimates with those given in this manuscript.
A7: This is a very good suggestion. We list the statistics of the chosen 8 spectral characteristic parameters in Table 4 of the revision.

Round 2
Reviewer 2 Report
Dear Authors
I have read the manuscript again. I am happy with the response to my comments. However, I have not seen the main text that refers to the new Table 4. Can you make sure where the Table 4 should be described in the main text?
Author Response
Reply to Reviewer 2 Comments
C1. Can you make sure where the Table 4 should be described in the main text?
A1.We thank the reviewer to point this. The new table 4 is in line 252 of the main text.
